# Factors Associated with Mortality in Nosocomial Lower Respiratory Tract Infections: An ENIRRI Analysis

**DOI:** 10.3390/antibiotics14020127

**Published:** 2025-01-26

**Authors:** Luis Felipe Reyes, Antoni Torres, Juan Olivella-Gomez, Elsa D. Ibáñez-Prada, Saad Nseir, Otavio T. Ranzani, Pedro Povoa, Emilio Diaz, Marcus J. Schultz, Alejandro H. Rodríguez, Cristian C. Serrano-Mayorga, Gennaro De Pascale, Paolo Navalesi, Szymon Skoczynski, Mariano Esperatti, Luis Miguel Coelho, Andrea Cortegiani, Stefano Aliberti, Anselmo Caricato, Helmut J. F. Salzer, Adrian Ceccato, Rok Civljak, Paolo Maurizio Soave, Charles-Edouard Luyt, Pervin Korkmaz Ekren, Fernando Rios, Joan Ramon Masclans, Judith Marin, Silvia Iglesias-Moles, Stefano Nava, Davide Chiumello, Lieuwe D. Bos, Antonio Artigas, Filipe Froes, David Grimaldi, Mauro Panigada, Fabio Silvio Taccone, Massimo Antonelli, Ignacio Martin-Loeches

**Affiliations:** 1Unisabana Center for Translational Science, School of Medicine, Universidad de La Sabana, Chia 250001, Colombia; luis.reyes5@unisabana.edu.co (L.F.R.);; 2Clinica Universidad de La Sabana, Chia 140013, Colombia; 3Pandemic Sciences Institute, University of Oxford, Oxford OX37LF, UK; 4School of Medicine, University of Barcelona, 08036 Barcelona, Spain; 5Instititut d´investigacions Biomédiques August Pi i Sunyer, 08036 Barcelona, Spain; 6Médecine Intensive-Réanimation, Hôpital R. Salengro, CHU de Lille, 59037 Lille, France; 7Université de Lille, CNRS, UMR 8576-UGSF-Unité de Glycobiologie Structurale et Fonctionnelle, 59000 Lille, France; 8Barcelona Institute for Global Health, ISGlobal, Hospital Clínic-Universitat de Barcelona, 08036 Barcelona, Spain; 9Pulmonary Division, Heart Institute (InCor), Hospital das Clinicas HCFMUSP, Faculdade de Medicina, Universidade de Sao Paulo, Sao Paulo 05508-220, Brazil; 10NOVA Medical School, NOVA University of Lisbon, 1169-056 Lisbon, Portugal; 11Center for Clinical Epidemiology and Research Unit of Clinical Epidemiology, OUH Odense University Hospital, 5230 Odense, Denmark; 12Intensive Care Unit 4, Department of Intensive Care, Hospital de São Francisco Xavier, CHLO, 1449-005 Lisbon, Portugal; 13School of Medicine, Corporació Sanitaria Parc Tauli, 08208 Sabadell, Spain; 14Departament de Medicina, Universitat Autonoma de Barcelona (UAB), 08193 Barcelona, Spain; 15Intensive Care, Amsterdam UMC, University of Amsterdam, 1105 Amsterdam, The Netherlands; 16Department of Intensive Care, Laboratory for Experimental Intensive Care & Anesthesiology (LEICA), 1105 Amsterdam, The Netherlands; 17Hospital Joan XXIII de Tarragona, 43003 Tarragona, Spain; ahr1161@yahoo.es; 18Engineering School, Universidad de La Sabana, Chia 111321, Colombia; 19Department of Intensive Care and Anesthesiology, Fondazione Policlinico Universitario A. Gemelli IRCCS, 00168 Rome, Italy; 20School of Medicine, Magna Graecia University, 88100 Catanzaro, Italy; 21Sant’Andrea (ASL VC), 13100 Vercelli, Italy; 22Department of Lung Diseases and Tuberculosis, Faculty of Medical Sciences in Zabrze, Medical University of Silesia, 41-803 Katowice, Poland; 23Hospital Privado de Comunidad, Escuela Superior de Medicina, Universidad Nacional de Mar del Plata, Mar del Plata 7600, Argentina; 24Department of Precision Medicine in Medical, Surgical and Critical Care Area (Me.Pre.C.C.), University of Palermo, 90127 Palermo, Italy; 25School of Medicine, Medical University of Silesia, 41-902 Katowise, Poland; 26Department of Biomedical Sciences, Humanitas University, 20072 Milan, Italy; 27Respiratory Unit, IRCCS Humanitas Research Hospital, 20089 Milan, Italy; 28Division of Infectious Diseases and Tropical Medicine, Department of Internal Medicine 4—Pneumology, Kepler University Hospital, 4020 Linz, Austria; 29Medical Faculty, Johannes Kepler University Linz, 4040 Linz, Austria; 30Ignaz Semmelweis Institute, Interuniversity Institute for Infection Research, 1090 Vienna, Austria; 31“Dr Fran Mihaljevic” University Hospital for Infectious Diseases, 10000 Zagreb, Croatia; rok.civljak@bfm.hr; 32Service de Médecine Intensive Réanimation, Sorbonne Université, Groupe Hospitalier Pitié-Salpêtriere, Assistance Publique–Hôpitaux de Paris, 75013 Paris, France; 33Medical Faculty, Ege University, 35100 Izmir, Turkey; 34Hospital Nacional Alejandro Posadas, El Palomar 1704, Argentina; 35Critical Care Department, Hospital del Mar, GREPAC, Hospital del Mar Research Institute, MELIS, Universitat Pompeu Fabra, 08003 Barcelona, Spain; 36Hospital del Mar, 08003 Barcelona, Spain; 37Hospital Arnau de Vilanova de Lleida, 25198 Lleida, Spain; 38Universita Alma Mater Studiorum Bologna Pneumologia e Terapia Intensiva Respiratoria, IRCCS Ospedale di Sant’Orsola, 40138 Bologna, Italy; 39Department of Medical and Surgical Sciences (DIMEC), University of Bologna, 40138 Bologna, Italy; 40ASST Santi Paolo e Carlo, 20142 Milan, Italy; 41Ntensive Care Medicine Department, Corporacion Sanitaria Universitaria Parc Tauli, Institut d’Investigació I Innovació Parc Tauli I3PT, CIBER Enfermedades Respiratorias, Autonomous University of Barcelona, 08208 Sabadell, Spain; 42Chest Department, Hospital Pulido Valente, CHULN, 1769-001 Lisbon, Portugal; 43Hopital Universitaire de Bruxelles (HUB), Université Libre de Bruxelles (ULB), 1050 Brussels, Belgium; 44Anesthesia and Critical Care, Fondazione IRCCS Ca’ Granda Ospedale Maggiore Policlinico, 20100 Milan, Italy; 45St James’s University Hospital, Trinity College, D08 NHY1 Dublin, Ireland

**Keywords:** critical care, mechanical ventilation, nosocomial lower respiratory tract infections

## Abstract

**Background**: Nosocomial lower respiratory tract infections (nLRTIs) are associated with unfavorable clinical outcomes and significant healthcare costs. nLRTIs include hospital-acquired pneumonia (HAP), ventilator-associated pneumonia (VAP), and other ICU-acquired pneumonia phenotypes. While risk factors for mortality in these infections are critical to guide preventive strategies, it remains unclear whether they vary based on their requirement of invasive mechanical ventilation (IMV) at any point during the hospitalization. **Objectives:** This study aims to identify risk factors associated with short- and long-term mortality in patients with nLRTIs, considering differences between those requiring IMV and those who do not. **Methods**: This multinational prospective cohort study included ICU-admitted patients diagnosed with nLRTI from 28 hospitals across 13 countries in Europe and South America between May 2016 and August 2019. Patients were selected based on predefined inclusion and exclusion criteria, and clinical data were collected from medical records. A random forest classifier determined the most optimal clustering strategy when comparing pneumonia site acquisition [ward or intensive care unit (ICU)] versus intensive mechanical ventilation (IMV) necessity at any point during hospitalization to enhance the accuracy and generalizability of the regression models. **Results**: A total of 1060 patients were included. The random forest classifier identified that the most efficient clustering strategy was based on ventilation necessity. In total, 76.4% of patients [810/1060] received IMV at some point during the hospitalization. Diabetes mellitus was identified to be associated with 28-day mortality in the non-IMV group (OR [IQR]: 2.96 [1.28–6.80], *p* = 0.01). The 90-day mortality-associated factor was MDRP infection (1.98 [1.13–3.44], *p* = 0.01). For ventilated patients, chronic liver disease was associated with 28-day mortality (2.38 [1.06–5.31] *p* = 0.03), with no variable showing statistical and clinical significance at 90 days. **Conclusions**: The risk factors associated with 28-day mortality differ from those linked to 90-day mortality. Additionally, these factors vary between patients receiving invasive mechanical ventilation and those in the non-invasive ventilation group. This underscores the necessity of tailoring therapeutic objectives and preventive strategies with a personalized approach.

## 1. Introduction

Nosocomial lower respiratory tract infections (nLRTI), encompasses hospital-acquired pneumonia (HAP), hospital-acquired pneumonia that requires ventilation (VHAP), ventilator-associated pneumonia (VAP), ventilator-associated tracheitis (VAT), and intensive care unit-acquired pneumonia that does not require ventilation (ICU-AP) [1]. HAP and VAP are two conditions in the latest international clinical guidelines [2,3]. For instance, HAP is often a less severe disease; however, up to 50% of patients can develop serious complications such as acute respiratory failure or even sepsis and septic shock that require management at the intensive care unit (ICU) [2,4]. On the other hand, up to 90% of pneumonia episodes in the ICU occur in patients undergoing (IMV), making this complication of ventilation a significant concern for clinicians [5,6]. Overall, patients who develop nosocomial pneumonia and require treatment in the ICU are at increased risk of mortality and additional burden on hospital stays and healthcare costs per patient, ranging from 10,000 to 40,000 USD [7].

Identifying risk factors to enhance patients’ clinical outcomes has become a priority [8]. Treatment approaches for nLRTIs typically involve broad-spectrum antibiotics targeting common pathogens such as *Acinetobacter baumannii*, *Klebsiella* spp., *Escherichia coli*, methicillin-resistant *Staphylococcus aureus* (MRSA), and resistant strains of *Pseudomonas aeruginosa* as described in the first analysis of this ENIRRI cohort [1]. Therapeutic empirical approaches include broad-spectrum antibiotics targeting prevalent pathogens depending on the local epidemiology; however, the increasing prevalence of multidrug-resistant pathogens (MDRP) poses significant challenges for effective therapy. Resistance mechanisms, including β-lactamase production, efflux pumps, and porin channel mutations, contribute to treatment failure and necessitate ongoing evaluation of antimicrobial efficacy [2,8,9,10]. This underscores the need for targeted empirical therapy based on local resistance patterns and pathogen sensitivity profiles. Novel antibiotics, such as β-lactam/β-lactamase inhibitor combinations and advanced carbapenems, show promise in overcoming resistance. However, it remains uncertain whether mortality risk factors differ between different phenotypes or subtypes of nLRTI such as those who are receiving IMV and those who are not.

Given the lack of sufficient evidence regarding the optimal method for grouping patients with nLRTIs and identifying factors associated with worse clinical outcomes, we proposed this large multinational study, the ENIRRI. Our study aims to determine the most effective way to categorize patients with nLRTIs and to evaluate the risk factors associated with mortality among these groups.

## 2. Results

A total of 1060 patients admitted to the ICU were included in the study, with the majority receiving IMV (76.4% [810/1060]) (Figure 1). Most of the patients enrolled were male, 72.5% (769/1060), with a mean age of 64 years across the entire cohort. The Random Forest classifier model found that the most optimal way of categorizing patients was into non-IMV and IMV groups, achieving an AUC ROC of 0.70 ± 0.02. The AUC graph and variables included alongside their feature importance can be found in the Appendix A. In contrast, classifying patients based on whether their condition was acquired in the ward or the ICU resulted in a lower performance, with an AUC ROC of 0.68 ± 0.03 (Appendix A).

The cohort was then divided into those not under invasive mechanical ventilation (non-IMV) (250/1060) and those under invasive mechanical ventilation (IMV) (810/1060). Non-IMV patients received various forms of respiratory support, distributed as follows: 72.4% (181/250) received supplementary oxygen, 13.6% (34/250) were managed with non-invasive positive pressure ventilation (NIPPV), and 14% (35/250) were treated with a high-flow nasal cannula (HFNC). Additionally, non-IMV patients also had a higher median [IQR] age (non-IMV: 66 [57–75] vs. IMV: 63 [49–73], *p* = 0.003) and had more comorbid conditions. The most frequent past medical condition for both non-IMV and IMV patients was a history of immunocompromise (non-IMV: 37.6% [94/250] vs. IMV: 21.0% [170/810], *p* < 0.001), chronic renal failure (non-IMV: 14.8% [37/250] vs. IMV: 10.2% [83/810], *p* = 0.05), and chronic heart disease (non-IMV: 30.8% [77/250] vs. IMV: 25.8% [209/810], *p* = 0.12). Furthermore, IMV patients presented more severe disease at nLRTI diagnosis based on median [IQR] SAPS II (non-IMV: 41 [30–54] vs. IMV: 48 [38–59], *p* < 0.001) and SOFA score (non-IMV: 6 [4–9] vs. IMV: 8 [5–10], *p* < 0.001). All the demographic characteristics are shown in Table 1.

Regarding nLRTI etiology, non-IMV patients developed MRSA infections more frequently (non-IMV: 8.4% [21/250] vs. IMV: 4.2% [34/810], *p* < 0.001), while IMV patients were more frequently infected by *Pseudomonas aeruginosa* (non-IMV: 6.4% [16/250] vs. IMV: 17.0% [138/810], *p* = 0.01). Also, IMV patients were infected in a higher percentage by MDR pathogens (non-IMV: 18.0% [45/250] vs. IMV: 26.8% [217/810], *p* = 0.01). Finally, ventilated patients had worse clinical outcomes based on the median [IQR] ICU length of stay (LOS) (non-IMV: 14 [7–25] vs. IMV: 22 [13–37], *p* < 0.001) and hospital LOS (non-IMV: 36 [23–61] vs. IMV: 39 [22–65], *p* = 0.38) as well as mortality at 28 days (non-IMV: 14.4% [36/250] vs. IMV: 20.6% [167/810], *p* = 0.03), and mortality at 90 days (non-IMV: 30.4% [76/250] vs. IMV: 34.9% [283/810], *p* = 0.19) (Table 1).

### 2.1. Mortality Analysis in the Whole Cohort

Logistic regression was performed to identify the risk factors associated with 28 d and 90 d mortality within the whole cohort presented in Appendix A, respectively. For both time points, a higher SAPS II score was an independent risk factor (28 d: 1.04 [1.03–1.05], *p* < 0.001; 90 d: 1.03 [1.02–1.04], *p* < 0.001). Notably, the need for IMV was only related to mortality at 28 days (28 d: 1.53 [1.01–2.32], *p* = 0.04; 90 d: 1.26 [0.91–1.75], *p* < 0.001). Finally, age was analyzed as a continuous variable, exposing a statistically significant relationship with mortality at both time points (28 d: 1.02 [1.00–1.03], *p* = 0.01; 90 d: 1.02 [1.01–1.03], *p* < 0.001). The model used had an appropriate fitness determined by the Hosmer–Lemeshow test of 0.15 for 28-day mortality and 0.58 for 90-day mortality.

### 2.2. Mortality Stratified by Ventilation Status

An adjusted logistic regression model was fitted for 28 d and 90 d mortality between non-IMV and IMV patients (Table 2, Table 3, Table 4 and Table 5). The risk factor that showed the strongest association with mortality in non-IMV patients at 28 days was diabetes mellitus (OR [95% CI]) (2.96 [1.28–6.80], *p* = 0.01) and the SAPS II score (1.05 [1.02–1.07], *p* < 0.01). At 90 days, they were the SAPS II score (1.04 [1.02–1.06], *p* < 0.01) and an MDRP agent causing pneumonia (1.98 [1.13–3.44] *p* = 0.01). The model used had an appropriate fitness determined by the Hosmer–Lemeshow test of 0.87 for 28 d mortality and 0.30 for 90 d mortality (Table 2 and Table 3).

Regarding the patients who required IMV (Table 4 and Table 5), a higher SAPS II score at nLRTI diagnosis (28 d: 1.04 [1.02–1.05], *p* < 0.01; 90 d: 1.03 [1.02–1.05], *p* < 0.01) and older age (28 d: 1.02 [1.01–1.04], *p* < 0.01; 90 d: 1.03 [1.01–1.04], *p* < 0.01) showed significant association with 28 d and 90 d mortality. Furthermore, chronic liver disease showed an association with 28-day mortality (2.38 [1.06–5.31], *p* = 0.03). Further information regarding the 90-day mortality model can be found in Table 5. Although included in this model, MRSA infections did not show a significant association at any point. The model used had an appropriate fitness determined by the Hosmer–Lemeshow test of 0.28 for 28-day mortality and 0.06 for 90-day mortality.

## 3. Discussion

This multicenter and multinational prospective cohort study is focused on patients with nLRTI admitted to the ICU. After a non-supervised statistical analysis, we found that the best way to cluster patients into comparable groups with nLRTI was by the requirement of invasive mechanical ventilation at any time point during their hospitalization. Notably, over two-thirds of patients required mechanical ventilation and had an increased 28-day mortality risk. Consequently, the study identified the risk factors associated with 28-day and 90-day mortality in non-IMV and IMV patients. Among non-IMV patients, diabetes mellitus and MDRP infection were found to be independent factors associated with 28 d and 90 d mortality when the model was adjusted by older age and severity at diagnosis using the SAPS II score. For IVM patients, chronic liver disease was found to be strongly associated with 28 d mortality when the model was adjusted by severity according to the SAPS II score and age.

nLRTI is commonly regarded as the most frequently acquired infection in the ICU [11,12]. It has been estimated that approximately 65% of nosocomial infections originate from respiratory sources [13]. These infections occur at a rate of 5 to 10 cases per 1000 hospital admissions in Europe and the United States [14,15,16]. In ventilated patients, it is of particular concern, as VAP accounts for 10 to 40% of ICU pneumonia cases. In some countries, it can reach over 90% of patients who are intubated and mechanically ventilated [14,15,16,17]. In our cohort, which included patients from Europe and South America, more than two-thirds of nLRTI cases were reported in ventilated patients, aligning with previous research findings. While Shah et al. have reported a 5% decrease in the incidence of VAP in the United States over the last decade [18], the overall prevalence has largely remained stable. Among patients with nLRTI, those who received IMV faced a significantly increased 28-day mortality risk compared to those without mechanical ventilation. The worldwide all-cause mortality rate associated with VAP falls within the range of 20% to 50% [14], which is consistent with our findings. Notably, our study is a pioneer in prospectively assessing the issue of nosocomial pneumonia in several countries in Europe and South America. By including a larger, more diverse patient population, we have reduced random error and addressed the demographic heterogeneity, enhancing our results’ robustness and generalizability. It remains crucial to identify risk predictors promptly to identify at-risk patients effectively.

Our study introduces a different perspective by comparing the risk factors between non-IMV and IMV patients. Among non-IMV patients, past medical conditions, microbial etiology, and systemic corticoids us in previous studies led to increase in short- and mid-term mortality. In 2023, E. Bouza et al. conducted a comparative study on the etiology of nosocomial bacteremic pneumonia in ventilated and non-ventilated patients over the past decade, revealing a higher prevalence of *S. aureus* and *P. aeruginosa* in HAP, with an increased mortality rate of more than 40% [19], a trend that is further confirmed by our findings, which present MDRP infections associated with higher mortality rates in non-IMV patients; this association is further supported by an observational study conducted by de Oliveira et al. [20] They found that MDRP infections were linked to higher mortality rates, longer hospital stays, and an elevated risk of requiring mechanical ventilation. These findings align with our observation that MDRP infections emerged as a significant risk factor at the 90-day mark. However, further research is needed to evaluate the impact of inappropriate empiric antibiotic therapy at admission, as well as antibiotic misuse, on the development of nLRTI caused by MDRPs [20].

Regarding comorbidities, diabetes mellitus (DM) has been identified as an independent factor associated with mortality in non-IMV patients. The observation that DM is associated with mortality at 28 days but not at 90 days can be explained by the impact of DM on exacerbating the acute phase of the infection. This chronic condition may not influence long-term mortality, as those who survive the initial phase are less likely to be affected by diabetes during the resolution phase of the illness. This aligns and is further supported by the findings from Equils et al. in a randomized controlled trial on methicillin-resistant *Staphylococcus aureus* (MRSA) nosocomial pneumonia, where they noted that diabetic patients had higher overall 28-day mortality rates compared to non-diabetic patients (23.5% vs. 14.7%; RD = 8.8%, 95% CI [1.4, 16.3]) but this trend did not persist in long term mortality [21]. Furthermore, Yakoub et al. described this in a 2023 longitudinal cohort study, that DM was a significant risk factor for mortality in nosocomial pneumonia (OR: 2.98, *p* = 0.004) [22]. Although both studies had a smaller sample size than our cohort, their findings align with our results.

In IVM patients, chronic hepatic disease was associated with 28-day mortality but not with 90-day mortality. Similarly to DM, this may be due to the significant impact of these chronic conditions during the acute phase of infection. Pasquale et al. in 2013 found that patients with liver disease had significantly higher 28-day and 90-day mortality rates (63% vs. 28%, *p* < 0.001; 72% vs. 38%, *p* < 0.001, respectively) compared to non-chronic liver disease patients. Although their results seem contradictory in terms of temporality to ours, their cohort only included patients who acquired pneumonia in the ICU. In contrast, our ventilated group included patients who were ventilated before developing an nLRTI and those who developed HAP after required ventilation. These differences in patient populations likely contributed to the variations in long-term outcomes between the two studies [23]. In the same vein, our results are further supported by Maruyama et al. [24] who found that chronic liver disease was a significant factor associated with 30-day mortality due to pneumonia from any cause, not specifically nosocomial acquired, with an (OR: 3.029, 95% CI [1.126–8.149], *p* = 0.028) in a sample size similar to ours. However, our study specifically found an association of chronic liver disease with mortality in IMV patients [24].

For IVM and non-IMV patients, the SAPS II score showed a statistically significant association with the evaluated outcome. SAPS II is a score that provides an estimate of the risk of death within the first 24 h from calculation without having to specify a primary diagnosis [16]; showing a discriminative power for ICU 24 h mortality in nosocomial pneumonia with an AUC [95% CI] 0.752 [0.656–0.848] and a specificity of 83.93 [25] when calculated on the day of nLRTI diagnosis, demonstrating that this score is rather valuable for the acute phase of the illness. While the SAPS II score was included in our regression models to determine associated factors at 28 and 90 days, it cannot be interpreted as a risk factor despite its statistical significance. The very low hazard ratio indicates that this variable primarily serves to adjust the model rather than to predict long-term mortality independently. Its discriminative power for mortality over extended periods is minimal, as supported by studies like Iwashyna et al., which showed that the predictive ability of acute illness severity scores, such as SAPS II, diminishes over time. Specifically, it found that while such scores accurately predict outcomes during ICU admission, their predictive power wanes after approximately 10 days in the ICU, giving way to other factors like age, sex, and chronic health status [26].

It is important to acknowledge the limitations of our study. First, using fixed data represents a significant limitation, particularly in predicting long-term outcomes, as it fails to capture the dynamic changes in patient health over time. Additionally, clustering patients into two groups, including specific subgroups like the IMV group, introduces bias, especially since patients with VHAP, whose nLRTI was not due to ventilation, were included. However, to our knowledge, this is a pioneering study strengthened by its multicentric and multinational scope across Europe and Latin America. Second, the evolving dynamics in pneumonia management, particularly the shift toward non-IMV strategies with high-flow nasal cannulas, further complicates the extrapolation of our results to current practice as the smaller sample was in the non-IMV group. A further limitation of this study is the lack of data on glucose control during hospitalization, as blood glucose measurements were not consistently collected as well as the lack of MELD score or Child-Pugh classification data for chronic liver disease. This prevented us from assessing their potential impact on patient outcomes and mortality in nLRTI.

Nevertheless, it is essential to mention that there is still limited migration to non-IMV strategies, especially in low- and middle-income countries, and the results found are valuable and provide a starting point for future research. Third, although DAGs suggest potential collinearity between variables like SAPS II and IMV, the statistical assessment shows minimal correlation (point-biserial correlation coefficient = −0.0765). Fourth, we could not assess the quality of sputum samples collected for microorganism identification, the techniques used for sample collection, the implementation of preventive strategies against infections caused by microorganisms other than *P. aeruginosa* and MRSA, or the specific antibiotic treatments administered. This information could have provided additional insights. However, it is essential to mention that even with high-quality sputum, the etiological agent of lower respiratory infections is only identified in 38% of all cases [27]. Fifth, the variability in the time taken to diagnose nosocomial lower respiratory tract infections (nLRTI) and the lack of standardized treatment protocols across different ICUs may have resulted in the omission of specific fungal and viral tests, potentially introducing bias. Although technical and diagnostic capabilities may be limited in some countries, guidelines for managing healthcare-associated respiratory infections are widely disseminated and applied globally, reducing bias. Despite these limitations, our results are valuable because they are consistent with previous studies and provide insights that can help physicians identify specific patient characteristics, ultimately leading to better patient care.

## 4. Materials and Methods

This multicenter, multinational prospective cohort study represents one of the primary analyses of the European network for ICU-related respiratory infections (ENIRRI) multicenter study across 28 ICUs in 13 countries throughout Europe and Latin America, including Argentina, Belgium, Colombia, Croatia, France, Germany, Ireland, Italy, the Netherlands, Poland, Portugal, Spain, and Turkey. The participating hospitals were selected based on logistical feasibility and their ability to contribute to the study objectives. The study enrolled critically ill patients admitted between 9 May 2016, and 16 August 2019, with each site conducting enrollment over a continuous 12-month period within this timeframe. Consecutive patients aged 18 years or older were included if they developed a lower respiratory tract infection (LRTI) at least 48 h after hospital admission (i.e., nosocomial LRTI), were later admitted to the ICU, and/or developed LRTI during their ICU stay. Follow-up was performed for all enrolled patients until hospital discharge.

The study followed the Code of Ethics of the World Medical Association (Declaration of Helsinki). The study received approval from the institution’s Internal Review Board (Comité Ètic d’Investigació Clínica, registry number HCB/2020/0370) and was registered in ClinicalTrials.gov Identifier (NCT03183921). Additionally, each of the thirteen participating sites obtained approval from its institutional ethics committee to conduct the study. Informed consent from patients was obtained when this was requested per local regulations. All clinical data were anonymized and transferred to the coordinating center for data curation and analysis. Further details are provided in reference [1].

### 4.1. Definitions

According to the 2016 ATS/IDS guidelines, pneumonia is characterized by new lung infiltrates accompanied by clinical indicators suggestive of an infectious origin, including recent onset of fever, purulent sputum, leukocytosis, and decline in oxygenation. Consequently, nosocomial pneumonia is diagnosed in patients who develop a pulmonary infection after being hospitalized for 48 h or more. HAP is identified explicitly as pneumonia occurring in patients after 48 h of being admitted to the hospital while not receiving invasive mechanical ventilation.

Acute kidney injury (AKI) was defined based on the Kidney Disease Improving Global Outcomes (KDIGO) classification, with a KDIGO Stage ≥ 2 [28]. In contrast, acute respiratory distress syndrome (ARDS) was defined according to the Berlin definition [29]. Multiorgan failure was determined when three or more organ systems failed following the diagnosis of nLRTI, and septic shock was defined as sepsis-induced hypotension with elevated lactate (≥2 mmol/L), persisting despite adequate fluid resuscitation [30,31].

### 4.2. Data Collection

All data were collected from the medical chart and transferred by the principal investigator to the multinational dataset. Demographics, type of admission, previous treatments, comorbidities, laboratories, Sequential Organ Failure Assessment (SOFA) score [32], new Simplified Acute Physiology Score (SAPS II) [33], clinical complications, microbiological information, and clinical outcomes, such as ICU length of stay (LOS), hospital LOS, 28-day mortality (28 d), and 90-day mortality (90 d), were included in the dataset. The microbiological diagnosis was confirmed by sputum in non-ventilated patients and using bronchoscopy or blind bronchoalveolar lavage (BAL) or tracheobronchial aspirates (TBA) in ventilated patients. The microbiological threshold was BAL ≥ 104 colony-forming units per mL and ≥105 colony-forming units for sputum or TBAs. Notably, microbiology assessment was performed based on international and local guidelines, not per study protocol. Ventilatory management strategies, treatments, and microbiological evaluations were not standardized across centers. Instead, these decisions were made at the discretion of the attending clinician, guided by local practices and supported by international recommendations.

### 4.3. Clustering Patients by Shared Demographic and Clinical Characteristics to Enhance Model Accuracy and Generalizability

After obtaining demographic, clinical, and laboratory features, missing data were addressed using the K-nearest neighbors (KNN) imputation method, applied only to variables with less than 30% missing data to maintain data integrity. Variables exceeding this threshold were excluded from the analysis to avoid introducing bias.

Patients were initially categorized into five groups as described in the first publication of this cohort: HAP, ventilated hospital-acquired pneumonia (VHAP), intensive care unit-acquired pneumonia (ICU-AP), VAP, and ventilator-associated tracheobronchitis (VAT) [1]. Two clustering strategies were proposed to enhance statistical power and identify phenotypically similar patient groups. The first strategy involved clustering patients based on the acquisition site of nLRTI (either in the general ward or the ICU), assuming that patients in the ICU were generally more critically ill. The second strategy was based on whether patients required mechanical ventilation at any point during their hospitalization, with the rationale that many patients were already ventilated before nLRTI development, often due to severe underlying conditions like trauma or major surgery. This clustering strategy aimed to reflect the severity of the patient’s pre-existing condition rather than the infection itself. Both methods were designed to create clusters of patients sharing similar clinical and demographic characteristics, optimizing the accuracy of risk factor identification.

Predictor variables were separated from the target variables, and a Random Forest Classifier model, known for its effectiveness with complex datasets and non-linear relationships, was chosen for training and clustering patients into groups that shared similar characteristics. The model underwent training using the Random Forest algorithm with 100 decision trees. Mechanical ventilation status and the place of acquisition, whether in the ward or ICU, were utilized as clinically plausible variables to cluster patients alongside all demographic variables.

Stratified K-fold cross-validation assessed model performance, revealing significant differences between ventilated and non-ventilated patient groups based on the selected variables. Model accuracy was evaluated using traditional accuracy metrics and the area under the receiver operating characteristic (AUROC) curve.

### 4.4. Statistical Analysis

All categorical variables are presented as relative and absolute frequencies and continuous variables with median (Interquartile Ranges [IQR]) or mean (Standard Deviations [SD]) depending on their distribution according to the Kolmogorov–Smirnov test. The non-normal variables were compared using the Mann–Whitney U and *t*-test for those with normal distribution. The x2 or Fisher’s exact test was used to compare categorical variables. The variables with 30% or more missing data were not used in the analysis to prevent bias and maintain integrity in the analysis.

First, a bivariate analysis was carried out to identify the factors related to the outcome. Bivariate analyses examine potential correlations between variables, guided by our directed acyclic graph (DAG) (Appendix A) to explore the relationships between covariates and mortality. Variables with a *p*-value < 0.2 in the bivariate analysis are considered for further inclusion in the multivariate logistic regression model. However, inclusion in the bivariate analysis does not guarantee inclusion in the final model.

The DAG is employed to identify potential collinear variables or those within others’ causal pathways, ensuring that only the most relevant variables are incorporated into the multivariate analysis. A backward elimination method is applied, with a stopping rule based on *p*-values. Variables with *p*-values between 0.3 and 0.4 are removed to maintain model parsimony, adjusting from the initially proposed threshold of 0.5 [34]. Combining bivariate analysis, DAG-guided selection, and backward elimination ensures robust and precise identification of risk factors.

The final model is adjusted based on the most significant and relevant covariates, ensuring appropriate adjustment and minimizing bias in estimating associations between risk factors and mortality. Then, multivariable logistic regression models assessed risk factors (HR [95% CI]) associated with 28 d and 90 d mortality among non-ventilated and ventilated patients. In the multivariate test, 28 and 90 days of mortality were taken as the dependent dichotomic variables. The non-IMV model was adjusted by age, taken as a dichotomic variable (>65 years), and severity using the SAPS II score as a continuous variable. The IMV model was adjusted by age and severity, and the SAPS II score was used in both cases as a continuous variable. The goodness of fitness of the multivariable logistic regression models was assessed with the Hosmer–Lemeshow test, and the level of significance considered at two-tailed was a *p*-value < 0.05. All the analyses were performed in the SPSS statistical package, version 29.

## 5. Conclusions

This multicenter, multinational study conducted in Europe and Latin America sheds light on the clinical landscape of nLRTI, which continues to be an essential issue in the context of critical care. We provide insights into the higher mortality risk associated with this condition and its risk factors, finding significant differences between non-invasive ventilated and invasive-ventilated patients in the critical care setting. These results highlight the necessity of moving beyond a one-size-fits-all approach, advocating instead for personalized strategies that consider patient severity, demographics, and the specific characteristics of the infection. Early identification of at-risk patients is essential for guiding targeted therapeutic efforts to improve outcomes. Future studies focusing on the distinct groups identified by the ENIRRI group could further refine risk stratification, enabling a more precise and personalized approach for these homogeneous yet distinct patient populations.

## Figures and Tables

**Figure 1 antibiotics-14-00127-f001:**
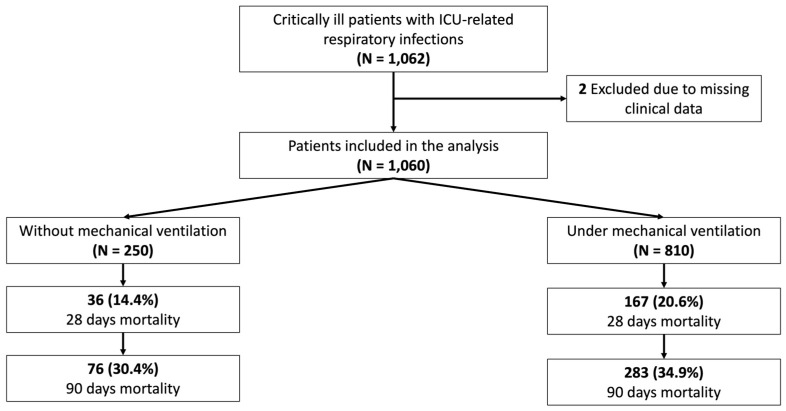
Study flowchart of included patients diagnosed with nLRTIs and clinical outcomes.

**Table 1 antibiotics-14-00127-t001:** Demographic and Clinical Characteristics of Patients with nLRTI stratified by those requiring IMV.

Variables	Invasive Mechanical Ventilation		*p*-Value
	No N = 250	Yes N = 810	
Demographics			
Gender (Male)	180 (72.0)	589 (72.7)	0.82
Age	66 [57–75]	63 [49–73]	0.003
>65 years	180 (72.0)	589 (72.7)	0.002
BMI	25.9 (22.3–29.4)	26.0 (23.3–29.4)	0.38
Comorbidities			
Diabetes Mellitus	48 (19.2)	167 (20.6)	0.62
Chronic Renal Disease	37 (14.8)	83 (10.2)	0.05
Immune Compromise	94 (37.6)	170 (21.0)	<0.001
Chronic Heart Disease	77 (30.8)	209 (25.8)	0.12
Chronic Liver Disease	17 (6.8)	49 (6.0)	0.83
Chronic Lung Disease	70 (28.0)	169 (20.9)	0.02
Drug Abuse	39 (15.6)	169 (20.9)	0.07
Severity at ICU admission			
SAPS II score	41.0 [30.0–54.0]	48.0 [38.0–59.0]	<0.001
SOFA score	6.0 [4.0–9.0]	8.0 [5.0–10.0]	<0.001
Systemic Corticoid Use	58 (23.2)	197 (24.3)	0.72
Coma	27 (10.8)	223 (27.5)	<0.001
Type of ICU admission			
Scheduled Surgery	31 (12.4)	64 (7.9)	<0.001
Emergency Surgery	24 (9.6)	139 (17.2)	
Medical	186 (74.4)	514 (63.5)	
Trauma	9 (3.6)	93 (11.5)	
Laboratory results on the first day of hospital admission			
Leucocytes	12.940 (8.150–17.700)	12.800 (9.600–17.470)	0.461
PaO^2^/FiO^2^	170.0 (126.0–242.4)	196.5 (140.0–266.0)	0.005
Complications			
Septic shock	70 (28.0)	256 (31.6)	0.28
Acute kidney injury	47 (18.8)	171 (21.1)	0.42
Multiorgan failure	29 (11.6)	130 (16.1)	0.08
Microbiological Etiology and Antibiotic Resistance			
MRSA infection	21 (8.4)	34 (4.2)	<0.001
*P. aeruginosa* infection	16 (6.4)	138 (17.0)	0.01
Recurrence infection	52 (20.8)	205 (25.3)	0.15
MDRP	45 (18.0)	217 (26.8)	0.01
Outcomes			
ICU LOS	14 [7–25]	22 [13–37]	<0.001
Hospital LOS	36 [23–61]	39 [22–65]	0.38
28 days mortality	36 (14.4)	167 (20.6)	0.03
90 days mortality	76 (30.4)	283 (34.9)	0.19

Data are presented as No. (%) or Median [IQR]. BMI: Body Mass Index, SAPS II: Simplified Acute Physiology Scores II, SOFA: Sequential Organ Failure Assessment, MRSA: Methicillin-Resistant *Staphylococcus aureus*, MDRP: Multidrug-Resistant Pathogens, ICU: Intensive Care Units, LOS: Length of Stay.

**Table 2 antibiotics-14-00127-t002:** Bivariate and multivariate analysis for 28 days mortality among non-IMV patients.

Variable	Bivariate		Multivariate	
	OR (95% IC)	*p*-Value	OR (95% IC)	*p*-Value
Demographic				
Age > 65	1.42 (0.62–3.30)	0.27	0.69 (0.30–1.52)	0.38
Gender (male)	1.43 (0.62–3.30)	0.55		
BMI	1.01 (0.95–1.06)	0.71		
Comorbidities				
Diabetes Mellitus	2.89 (1.34–6.25)	0.01	2.96 (1.28–6.80)	0.01
Chronic Renal Disease	2.28 (0.45–3.07)	0.80		
Immunocompromised Disease	0.80 (0.38–1.70)	0.71		
Chronic Heart Disease	1.15 (0.54–2.43)	0.70		
Chronic Liver Disease	0.78 (0.17–3.57)	1.00		
Lung Disease	1.34 (0.63–2.86)	0.43		
Drug Abuse	1.37 (0.55–3.40)	0.46		
Severity				
SAPS II (Pneumonia Diagnosis)	1.04 (1.02–1.06)	<0.001	1.05 (1.02–1.07)	<0.001
Systemic Corticoid Use	1.56 (0.72–3.41)	0.29		
Coma	0.72 (0.20–2.53)	0.78		
Complications				
Acute kidney injury	2.56 (1.17–5.59)	0.02		
Microorganism and Antibiotic-Resistance Pattern				
*P. aeruginosa* infection	2.64 (0.74–9.41)	0.13		
MRSA infection	1.15 (0.30–4.43)	0.73		
MDRP	0.70 (0.26–1.92)	0.64	1.62 (0.83–3.18)	0.15

BMI: Body Mass Index, SAPS II: Simplified Acute Physiology Score II, MRSA: Methicillin-Resistant *Staphylococcus aureus*, MDRP: Multidrug-Resistant Pathogens.

**Table 3 antibiotics-14-00127-t003:** Bivariate and multivariate analysis for 90-day mortality among non-IMV patients.

Variable	Bivariate		Multivariate	
	OR (95% IC)	*p*-Value	OR (95% IC)	*p*-Value
Demographic				
Age > 65 yrs	1.10 (0.92–1.32)	0.16	0.91 (0.48–1.73)	0.77
Gender (male)	1.43 (0.62–3.30)	0.55		
BMI	0.99 (0.95–1.03)	0.71		
Comorbidities				
Diabetes Mellitus	2.89 (1.34–6.25)	0.01	1.37 (0.67–2.80)	0.37
Chronic Renal Disease	1.18 (0.45–3.07)	0.80		
Immunocompromised Disease	0.80 (0.38–1.70)	0.71		
Chronic Heart Disease	1.15 (0.54–2.43)	0.70		
Chronic Liver Disease	0.78 (0.17–3.57)	1.00		
Lung Disease	1.34 (0.63–2.86)	0.43		
Drug Abuse	1.37 (0.55–3.40)	0.46		
Severity				
SAPS II (Pneumonia Diagnosis)	1.04 (1.03–1.06)	<0.001	1.04 (1.02–1.06)	<0.001
Systemic Corticoid Use	1.56 (0.72–3.41)	0.29		
Coma	0.72 (0.20–2.53)	0.78		
Complications				
Acute kidney injury	2.56 (1.17–5.59)	0.02		
Microorganism and Antibiotic-Resistance Pattern				
*P. aeruginosa* infection	2.64 (0.74–9.41)	0.13		
MRSA infection	1.15 (0.30–4.43)	0.73		
MDRP	0.70 (0.26–1.92)	0.64	1.98 (1.13–3.44)	0.01

BMI: Body Mass Index, SAPS II: Simplified Acute Physiology Score II, MRSA: Methicillin-Resistant *Staphylococcus aureus*, MDRP: Multidrug-Resistant Pathogens.

**Table 4 antibiotics-14-00127-t004:** Bivariate and multivariate analysis for 28 days mortality among patients under IMV.

Variable	Bivariate		Multivariate	
	OR (95% IC)	*p*-Value	OR (95% IC)	*p*-Value
Demographic				
Age	1.03 (1.02–1.04)	<0.001	1.02 (1.01–1.04)	<0.01
Gender (male)	1.10 (0.75–1.63)	0.70		
BMI	1.02 (0.99–1.05)	0.18		
Comorbidities				
Diabetes Mellitus	1.03 (0.68–1.56)	0.91		
Chronic Renal Disease	1.66 (1.00–2.76)	0.06		
Immunocompromised Disease	0.95 (0.63–1.45)	0.92		
Chronic Heart Disease	1.35 (0.93–1.96)	0.14		
Chronic Liver Disease	1.96 (1.05–3.65)	0.04	2.38 (1.06–5.31)	0.03
Lung Disease	1.25 (0.84–1.88)	0.29		
Drug Abuse	0.68 (0.43–1.07)	0.11		
Severity				
SAPS II (Pneumonia Diagnosis)	1.04 (1.03–1.06)	<0.001	1.04 (1.02–1.05)	<0.001
Systemic Corticoid Use	0.94 (0.63–1.40)	0.84		
Coma	0.93 (0.63–1.36)	0.77		
Complications				
Acute kidney injury	1.23 (0.82–1.84)	0.34		
Microorganism and Antibiotic-Resistance Pattern				
*P. aeruginosa* infection	0.59 (0.34–1.01)	0.06		
MRSA infection	1.90 (0.88–4.10)	0.11	2.02 (0.83–4.9)	0.11
MDRP	0.86 (0.58–1.28)	0.49		

BMI: Body Mass Index, SAPS II: Simplified Acute Physiology Score II, MRSA: Methicillin-Resistant *Staphylococcus aureus*, MDRP: Multidrug-Resistant Pathogens.

**Table 5 antibiotics-14-00127-t005:** Bivariate and multivariate analysis for 90 days mortality among patients under IMV.

Variable	Bivariate		Multivariate	
	OR (95% IC)	*p*-Value	OR (95% IC)	*p*-Value
Demographic				
Age	1.03 (1.02–1.04)	<0.001	1.03 (1.01–1.04)	<0.001
Gender (male)	1.03 (0.75–1.43)	0.87		
BMI	1.02 (0.99–1.04)	0.30		
Comorbidities				
Diabetes Mellitus	1.46 (1.03–2.06)	0.04		
Chronic Renal Disease	2.18 (1.38–3.45)	0.001		
Immunocompromised Disease	1.32 (0.93–1.87)	0.13		
Chronic Heart Disease	1.73 (1.26–2.39)	0.001		
Chronic Liver Disease	1.71 (0.95–3.05)	0.09	1.94 (0.91–4.10)	0.08
Lung Disease	1.21 (0.85–1.72)	0.28		
Drug Abuse	0.79 (0.55–1.14)	0.21		
Severity				
SAPS II (Pneumonia Diagnosis)	1.04 (1.03–1.05)	<0.001	1.03 (1.02–1.05)	<0.001
Systemic Corticoid Use	1.31 (0.94–1.82)	0.12		
Coma	1.18 (0.86–1.62)	0.32		
Complications				
Acute kidney injury	1.62 (1.15–2.28)	0.01		
Microorganism and Antibiotic-Resistance Pattern				
*P. aeruginosa* infection	0.71 (0.47–1.08)	0.12		
MRSA infection	1.59 (0.79–3.20)	0.20	1.60 (0.70–3.64)	0.26
MDRP	1.25 (0.91–1.73)	0.18		

BMI: Body Mass Index, SAPS II: Simplified Acute Physiology Score II, MRSA: Methicillin-Resistant *Staphylococcus aureus*, MDRP: Multidrug-Resistant Pathogens.

## Data Availability

Data and materials will be available upon request to the corresponding author.

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
