# Peer review of "Factors Associated with Mortality in Nosocomial Lower Respiratory Tract Infections: An ENIRRI Analysis"

_antibiotics, 2025, doi:10.3390/antibiotics14020127_

Round 1
Reviewer 1 Report
Comments and Suggestions for Authors
Dear Authors,
I sincerely thank you for the opportunity to review this manuscript, that I read with great interest. I only found a little typo in Line 96, where I suggest to replace "which encompasses" with "encompass".
Apart from this detail, I express my congratulations to your overall research group for the precision and scientific soundness of this multiceter and multinational study. I did not detect any methodological or content issue and thus I do really consider this work as a valuable resource in our research field in its present form.
I remain at your disposal.
Best Regards
Author Response
Comment 1: I sincerely thank you for the opportunity to review this manuscript, that I read with great interest. I only found a little typo in Line 96, where I suggest to replace "which encompasses" with "encompass".
Apart from this detail, I express my congratulations to your overall research group for the precision and scientific soundness of this multiceter and multinational study. I did not detect any methodological or content issue and thus I do really consider this work as a valuable resource in our research field in its present form.
I remain at your disposal.
Response:
We sincerely thank the reviewer for their valuable comments on this research work. We firmly believe that the analysis presented here significantly enhances our understanding of how different nLRTI phenotypes vary in terms of presentation, severity, and risk factors for unfavorable outcomes, such as mortality. This study contributes to our ability to identify at-risk populations earlier and enables a more personalized and specific approach to prevention and closer attention to those at risk. Such an approach fosters holistic decision making rather than relying on generalized prevention and treatment strategies, which may overlook critical details that impact patient outcomes.
Additionally, we have addressed the typographical error identified by the reviewer. The corrected text now reads as follows:
"Nosocomial lower respiratory tract infections (nLRTI) encompass hospital-acquired pneumonia (HAP), hospital-acquired pneumonia requiring ventilation (VHAP), ventilator-associated pneumonia (VAP), ventilator-associated tracheitis (VAT), and intensive care unit-acquired pneumonia that does not require ventilation (ICU-AP) [1]."
Thank you for taking the time to review this work.
Reviewer 2 Report
Comments and Suggestions for Authors
Title; Factors associated with mortality in Nosocomial Lower Respiratory Tract Infections: An ENIRRI analysis. (Manuscript ID - antibiotics-3366166)
Dear Editor,
Thank you for inviting me to review this original article.
I found the work to be original and believe it fills significant gaps in the field of antibiotic treatment outcomes.
In the abstract section, the background should be clearly separated from the objective. The background should provide a concise summary of the overall study, while the objective should outline the general purpose of the research.
The total number of participants, selection procedures, and data acquisition processes should be briefly mentioned in the methods portion of the summary section.
In the introduction section, the authors mentioned certain pathogens responsible for NLRTI. Are these listed pathogens the sole cause across all hospital setups? If not, how can the authors generalize the results of this study to other settings?
Additionally, the introduction should address treatment approaches, the potential antibiotics acting on NLRTI, their resistance profiles, and future plans to overcome this issue. It would also be beneficial to discuss the sensitivity of the pathogens, any resistance mechanisms, and the specificity of the antimicrobial for the pathogens in the introduction section.
It is stated that intravenous antibiotic use within 90 days prior to hospital admission can increase the occurrence of MDRP in VAP/HAP-related infections, including MRSA and resistant strains of P. aeruginosa. It would be better to clarify the role of prior antibiotic exposure in the development of MDRP pathogens.
The Materials and Methods section appears to contain content that would be more appropriately placed in the introduction.
In the methodology section, it is noted that 28 hospitals across European and South American countries were included. However, the authors should clarify how these hospitals were selected and how the study participants were recruited.
The data collection procedure and the tools used for data collection are not clearly described. These details should be explicitly provided in the methods section.
Lastly, the conclusion section should be placed immediately after the discussion section.
Author Response
Comment 1: Thank you for inviting me to review this original article.
I found the work to be original and believe it fills significant gaps in the field of antibiotic treatment outcomes.
Response: We sincerely thank the reviewer for taking the time to evaluate this research and project. We believe the outcomes and analyses presented are relevant to advancing the field, as they underscore the need for interventions beyond antibiotic treatment to achieve favorable outcomes. By identifying the risk factors associated with different nLRTI phenotypes, this study enables clinicians to make informed decisions regarding preventive strategies for poor outcomes, such as mortality. It also highlights the importance of closely monitoring at-risk populations and considering adjuvant therapies to improve outcomes.
This personalized approach represents a shift from generalized management strategies, emphasizing tailored care to enhance patient recovery.
Thank you again for reviewing our work. We remain available for further discussions that could enrich our collective understanding of this critical topic.
Comment 2: In the abstract section, the background should be clearly separated from the objective. The background should provide a concise summary of the overall study, while the objective should outline the general purpose of the research.
Response: We sincerely thank the reviewer for their insightful comment. We have carefully revised the abstract to ensure a clear separation between the background and objectives. In doing so, we have provided a concise summary of the study in the background section while clearly outlining the general purpose of the research in the objectives. The revised abstract now reads:
“Background: Nosocomial lower respiratory tract infections (nLRTIs) are associated with unfavorable clinical outcomes and significant healthcare costs. nLRTIs include hospital-acquired pneumonia (HAP), ventilator-associated pneumonia (VAP), and other ICU-acquired pneumonia phenotypes. While risk factors for mortality in these infections are critical to guide preventive strategies, it remains unclear whether they vary based on their requirement of invasive mechanical ventilation (IMV) at any point during the hospitalization.
Objectives: This study aims to identify risk factors associated with short- and long-term mortality in patients with nLRTIs, considering differences between those requiring IMV and those who do not.”
Comment 3: The total number of participants, selection procedures, and data acquisition processes should be briefly mentioned in the methods portion of the summary section.
Response: We thank the reviewer for highlighting this oversight. We have now revised the methods section of the abstract to include details about the selection procedures and data acquisition process, ensuring a clearer description. The total number of participants is mentioned in the results section. The updated methods section now reads:
“Methods: This multinational prospective cohort study included ICU-admitted patients diag-nosed with nLRTI from 28 hospitals across 13 countries in Europe and South America between May 2016 and August 2019. Patients were selected based on predefined inclusion and exclusion criteria, and clinical data were collected from medical records. A random forest classifier de-termined the most optimal clustering strategy when comparing pneumonia site acquisition [ward or intensive care unit (ICU)] versus ventilation necessity at any point during hospitalization to enhance the accuracy and generalizability of the regression models.”
Comment 4: In the introduction section, the authors mentioned certain pathogens responsible for NLRTI. Are these listed pathogens the sole cause across all hospital setups? If not, how can the authors generalize the results of this study to other settings?
Response: We thank the reviewer for this insightful comment, as we recognize the importance of improving the clarity of this section to avoid any potential confusion. This study does not specifically focus on nLRTIs caused solely by Pseudomonas aeruginosa or Staphylococcus aureus with resistance patterns. In the initial phase of this research, we comprehensively described the microbiological etiologies of nLRTIs where detection was possible. The findings highlighted the prevalence of pathogens such as Acinetobacter spp., Klebsiella spp., and Escherichia coli, among others. These pathogens are detailed across the five phenotypes of nLRTIs described in this study. To enhance understanding and provide a clearer expression of this idea, we have revised the text to explicitly mention these pathogens, concluding with "among others," as described in the descriptive phase of this study. Additionally, we have included a reference to the initial descriptive phase to ensure context and continuity.
The section now reads “Treatment approaches for nLRTIs typically involve broad-spectrum antibiotics targeting common pathogens such as Acinetobacter baumannii, Klebsiella spp., Escherichia coli, Methicillin-resistant Staphylococcus aureus (MRSA), and resistant strains of Pseudomonas aeruginosa as described in the first analysis of this ENIRRI cohort [1].
Comment 5: Additionally, the introduction should address treatment approaches, the potential antibiotics acting on NLRTI, their resistance profiles, and future plans to overcome this issue. It would also be beneficial to discuss the sensitivity of the pathogens, any resistance mechanisms, and the specificity of the antimicrobial for the pathogens in the introduction section.
Response:
Thank you for highlighting this important aspect for our introduction. After discussing with the co-authors, we agree that providing the reader with more context is crucial for understanding the relevance of this study.
The section now reads “Therapeutic empirical approaches include broad-spectrum antibiotics targeting prevalent pathogens depending the local epidemiology, however, the increasing prevalence of multidrug-resistant pathogens (MDRP) poses significant challenges for effective therapy. Resistance mechanisms, including β-lactamase production, efflux pumps, and porin channel mutations, contribute to treatment failure and necessitate ongoing evaluation of antimicrobial efficacy [2, 8-10]. This underscores the need for targeted empirical therapy based on local resistance patterns and pathogen sensitivity profiles. Novel antibiotics, such as β-lactam/β-lactamase inhibitor combinations and advanced carbapenems, show promise in overcoming resistance. However, it remains uncertain whether mortality risk factors differ between different phenotypes or subtypes of nLRTI such as those who are receiving IMV and those who are not.”
Comment 6: It is stated that intravenous antibiotic use within 90 days prior to hospital admission can increase the occurrence of MDRP in VAP/HAP-related infections, including MRSA and resistant strains of P. aeruginosa. It would be better to clarify the role of prior antibiotic exposure in the development of MDRP pathogens.
Response: We sincerely thank you for this insightful comment. As mentioned earlier, the reference to S. aureus and P. aeruginosa was not intended to focus exclusively on these two specific microbiological agents. Instead, these were provided as examples of microorganisms that have gained particular relevance in the context of nLRTI. Moreover, it is well established that the misuse of antibiotic therapy can increase the risk of microorganisms, such as bacteria, developing new resistance mechanisms against emerging antibiotic agents. However, this issue, while critical, is beyond the primary scope of our study.
Our main objective was to identify mortality risk factors within subgroups of nLRTI, including the different clinical phenotypes, as assessed in the initial analysis of the ENIRRI cohort. Specifically, we aim to determine whether these risk factors vary over time and between subgroups. As presented in our study's results, MDRP has emerged as a significant risk factor. While the implications of prior antibiotic use on the emergence or prevalence of MDRP pathogens warrant further investigation, we believe this is best addressed in a separate study.
We have, however, acknowledged this important point in the discussion section, which now states: “These findings align with our observation that MDRP infections emerged as a significant risk factor at the 90-day mark. However, further research is needed to evaluate the impact of inappropriate empiric antibiotic therapy at admission, as well as antibiotic misuse, on the development of nLRTI caused by MDRP pathogens [27”
Comment 7: The Materials and Methods section appears to contain content that would be more appropriately placed in the introduction.
Response: We thank the reviewer for their valuable comment on this aspect. The different phenotypes identified in the methods definition section have been summarized and incorporated into the introduction to provide readers with a clearer understanding that the analysis will be based on these phenotypes hence why this information was removed from the methods section. Beyond this adjustment, we believe the information presented in the methods section offers sufficient and relevant details for the reader.
Comment 8: In the methodology section, it is noted that 28 hospitals across European and South American countries were included. However, the authors should clarify how these hospitals were selected and how the study participants were recruited.
Response: We sincerely thank the reviewer for highlighting this important point. To address the concern, we have clarified the criteria for selecting participating hospitals and provided additional details regarding patient recruitment and follow-up, which were previously omitted. The revised section now reads as follows:
“This multicenter, multinational prospective cohort study is part of the European Network for ICU-Related Respiratory Infections (ENIRRI) initiative and represents one of its primary analyses. The study was conducted across 28 ICUs in 13 countries spanning Europe and Latin America, including Argentina, Belgium, Colombia, Croatia, France, Germany, Ireland, Italy, the Netherlands, Poland, Portugal, Spain, and Turkey. The participating hospitals were selected based on logistical feasibility and their ability to contribute to the study objectives. The study enrolled critically ill patients admitted between May 9, 2016, and August 16, 2019, with each site conducting enrollment over a continuous 12-month period within this timeframe. Consecutive patients aged 18 years or older were included if they developed a lower respiratory tract infection (LRTI) at least 48 hours after hospital admission (i.e., nosocomial LRTI), were later admitted to the ICU, and/or developed LRTI during their ICU stay. Follow-up was performed for all enrolled patients until hospital discharge.”
Comment 9: The data collection procedure and the tools used for data collection are not clearly described. These details should be explicitly provided in the methods section.
Response: We thank the reviewer for highlighting this important point. The data collection section of the methodology provides details about the tools used and references the specific data collected for this study. However, we recognized that the process for capturing ventilatory data was not clearly outlined. To address this, we have clarified this aspect in the revised section, which now states:
"Ventilatory management strategies, treatments, and microbiological evaluations were not standardized across centers. These decisions were made at the discretion of the attending clinician, guided by local practices and supported by international recommendations. Ventilatory data, including type and duration of support, were systematically recorded as part of the collected dataset to ensure consistency and completeness."
Comment 10: Lastly, the conclusion section should be placed immediately after the discussion section.
Response: Thank you for this valuable recommendation. We have implemented your suggestion, as we believe it enhances the clarity and flow of the manuscript. Accordingly, the conclusion section has been moved to the end of the discussion to improve readability and logical structure.
Reviewer 3 Report
Comments and Suggestions for Authors
This is an excellent paper. The authors could consider some minor changes:
1. Line 84: please add “intensive mechanical ventilation” and brackets for IMV
2. Line 111 and 211: the MDRPs should be listed since intensive care units have often more MDRPs than MRSA and MDR Pseudomonas bacteria. Acineto, KPC, ESBL and AmpC could be written down and, if percentages are available, it could be interesting to see them in a table.
3. Line 174: diabetes mellitus is found to be a risk factor but it is not stated the level of glucose control at admission and during hospital stay
4. Line 190: Chronic liver disease is another risk factor, but is not given a Meld score or a Child-Pough classification when it is known that the chronic liver disease stage is an important prognostic factor.
5. Line 461, 466 e 470: Methodology is duplicated 3 times; please delete the duplications
6. Supplement material: DAG for logistic regression could be better explained since it is not often used. Even the Random Forrest classification is not commonly used so that most readers could appreciate some brief information in comparison with other statistical methods.
Finally, if Antibiotics journal accepts coloured tables without additional charges, the authors could consider reformatting their tables using one or two light colours.
Author Response
Comment 1: This is an excellent paper. The authors could consider some minor changes:
Response: We thank the reviewer for taking the time to review our study and for the thoughtful and relevant comments.
Comment 2: Line 84: please add “intensive mechanical ventilation” and brackets for IMV
Response: We sincerely thank the reviewer for highlighting this oversight, as it clearly enhances the readability and understanding of this section. We have taken the reviewer’s recommendation into account and have incorporated "intensive mechanical ventilation (IMV)" into the respective line.
Comment 3. Line 111 and 211: the MDRPs should be listed since intensive care units have often more MDRPs than MRSA and MDR Pseudomonas bacteria. Acineto, KPC, ESBL and AmpC could be written down and, if percentages are available, it could be interesting to see them in a table.
Response: We thank the reviewer for this insightful comment, which was also noted by other reviewers. Indeed, this is an important point that we initially overlooked in the introduction and background, potentially leading to confusion or misinterpretation among readers. In the initial analysis of the ENIRRI cohort, a comprehensive description of the microorganisms responsible for nLRTI was conducted. These findings have now been incorporated into the background section, along with a citation of the first ENIRRI paper to provide additional context and clarity. This addition enhances the overall understanding of the microbiological landscape in ICUs.
This section now reads: “Treatment approaches for nLRTIs typically involve broad-spectrum antibiotics targeting common pathogens such as Acinetobacter baumannii, Klebsiella spp., Escherichia coli, Methicillin-resistant Staphylococcus aureus (MRSA), and resistant strains of Pseudomonas aeruginosa as described in the first analysis of this ENIRRI cohort [1].”
Comment 4: Line 174: diabetes mellitus is found to be a risk factor but it is not stated the level of glucose control at admission and during hospital stay
Response: We sincerely thank the reviewer for this insightful comment. Unfortunately, data regarding the level of glucose control at admission and during the hospital stay was not available, as blood glucose measurements were not systematically collected throughout the hospitalization. Therefore, we were unable to assess the impact of glucose control on patient outcomes in this study. However we have included this in the limitations section that now reads:
“A further limitation of this study is the lack of data on glucose control during hospitali-zation, as blood glucose measurements were not consistently collected as well as the lack of MELD score or Child-Pugh classification data for chronic liver disease. This prevented us from assessing their potential impact on patient outcomes and mortality in nLRTI.”
Comment 5: Line 190: Chronic liver disease is another risk factor, but is not given a Meld score or a Child-Pough classification when it is known that the chronic liver disease stage is an important prognostic factor.
Response: We appreciate the reviewer’s insightful comment. Unfortunately, we were unable to obtain the MELD score or Child-Pugh classification for patients with chronic liver disease, as this information was not systematically collected during the study. We acknowledge that these scoring systems are important prognostic factors and agree that their inclusion would have provided a more comprehensive understanding of the impact of chronic liver disease on patient outcomes. We have incorporated a section on the limitations section of the study that now reads:
“A further limitation of this study is the lack of data on glucose control during hospitali-zation, as blood glucose measurements were not consistently collected as well as the lack of MELD score or Child-Pugh classification data for chronic liver disease. This prevented us from assessing their potential impact on patient outcomes and mortality in nLRTI.”
Comment 6: Line 461, 466 e 470: Methodology is duplicated 3 times; please delete the duplications
Response: We thank the reviewer for this comment. We have removed the duplicate occurrences of the term "methodology," and the text now reads more clearly and concisely.
Comment 7: Supplement material: DAG for logistic regression could be better explained since it is not often used. Even the Random Forrest classification is not commonly used so that most readers could appreciate some brief information in comparison with other statistical methods.
Response: Thank you for the comment. We have added a section explaining the use of DAG and random forest to help readers better understand their role in enhancing the results and validity of our study. The revised section now reads:
“The DAG is employed to identify potential collinear variables or those within others' causal pathways, ensuring that only the most relevant variables are incorporated into the multivariate analysis. A backward elimination method is applied, with a stopping rule based on p-values. Variables with p-values between 0.3 and 0.4 are removed to maintain model parsimony, adjusting from the initially proposed threshold of 0.5[17]. Combining bivariate analysis, DAG-guided selection, and backward elimination ensures robust and precise identification of risk factors.”
Comment 8: Finally, if Antibiotics journal accepts coloured tables without additional charges, the authors could consider reformatting their tables using one or two light colours.
Response: Thank you for this thoughtful suggestion. While we appreciate the potential for enhanced visual appeal with the use of colours, however we have formatted the tables in accordance with the guidelines provided by the journal.
Round 2
Reviewer 2 Report
Comments and Suggestions for Authors
Dear authors, thanks for addressing my comments and revising the manuscripts accordingly. The revised manuscript addressed all my concerns and can be accepted at this stage.
Comments on the Quality of English LanguageEven if the manuscript was amended and revised, it still benefits from language revision and modification to make it more understandable for readers and convey the information it generates easily.